# Ethyl Caffeate Can Inhibit Aryl Hydrocarbon Receptor (AhR) Signaling and AhR-Mediated Potentiation of Mast Cell Activation

**DOI:** 10.3390/ijms24129997

**Published:** 2023-06-10

**Authors:** Phuc-Tan Nguyen, Yuki Nakamura, Nguyen Quoc Vuong Tran, Kayoko Ishimaru, Thuy-An Nguyen, Yoshiaki Kobayashi, Fumie Watanabe-Saito, Tohru Okuda, Nobuhiro Nakano, Atsuhito Nakao

**Affiliations:** 1Department of Immunology, Faculty of Medicine, University of Yamanashi, Yamanashi 409-3898, Japan; g20ddm09@yamanashi.ac.jp (P.-T.N.); vtran@yamanashi.ac.jp (N.Q.V.T.); ynakamura@yamanashi.ac.jp (Y.N.); ikayoko@yamanashi.ac.jp (K.I.); g20ddm08@yamanashi.ac.jp (T.-A.N.); yoshiakik@yamanashi.ac.jp (Y.K.); 2The Institute of Enology and Viticulture, University of Yamanashi, Yamanashi 400-0005, Japan; fumies@yamanashi.ac.jp (F.W.-S.); okuda@yamanashi.ac.jp (T.O.); 3Atopy Research Center, Juntendo University School of Medicine, Tokyo 113-8421, Japan; nbnakano@juntendo.ac.jp; 4Yamanashi GLIA Center, University of Yamanashi, Yamanashi 409-3898, Japan

**Keywords:** ethyl caffeate, aryl hydrocarbon receptor, mast cell, IgE, allergy

## Abstract

Ethyl caffeate (EC) is a natural phenolic compound that is present in several medicinal plants used to treat inflammatory disorders. However, its anti-inflammatory mechanisms are not fully understood. Here, we report that EC inhibits aryl hydrocarbon receptor (AhR) signaling and that this is associated with its anti-allergic activity. EC inhibited AhR activation, induced by the AhR ligands FICZ and DHNA in AhR signaling-reporter cells and mouse bone marrow-derived mast cells (BMMCs), as assessed by AhR target gene expressions such as *CYP1A1*. EC also inhibited the FICZ-induced downregulation of AhR expression and DHNA-induced IL-6 production in BMMCs. Furthermore, the pretreatment of mice with orally administered EC inhibited DHNA-induced *CYP1A1* expression in the intestine. Notably, both EC and CH-223191, a well-established AhR antagonist, inhibited IgE-mediated degranulation in BMMCs grown in a cell culture medium containing significant amounts of AhR ligands. Furthermore, oral administration of EC or CH-223191 to mice inhibited the PCA reaction associated with the suppression of constitutive *CYP1A1* expression within the skin. Collectively, EC inhibited AhR signaling and AhR-mediated potentiation of mast cell activation due to the intrinsic AhR activity in both the culture medium and normal mouse skin. Given the AhR control of inflammation, these findings suggest a novel mechanism for the anti-inflammatory activity of EC.

## 1. Introduction

Ethyl caffeate (ethyl (*E*)-3-(3,4-dihydroxy phenyl)prop-2-enoate or caffeic acid ethyl ester) (EC) is an ester of the hydroxycinnamic acid group (Figure 1A) found in several medicinal plants, including *Bidens pilosa*, which are used to treat inflammatory disorders [1]. EC has been shown to have several anti-inflammatory activities that may underlie the beneficial effects of some traditional medicines. For instance, EC suppresses lipopolysaccharide-induced nitric oxide production, which is associated with the inhibition of NF-κβ activation in vitro and exerts antiarthritic effects on collagen-induced arthritis by inhibiting the interferon-γ response in mice [2,3]. However, it is not fully understood how EC mediates these effects or whether any other mechanisms are involved in the anti-inflammatory activity of EC.

The aryl hydrocarbon receptor (AhR) is a chemical sensor for many common environmental contaminants (e.g., tetrachiorodibenzo-p-dioxin: TCDD), polycystic aromatic hydrocarbons, dietary components, microbial metabolites (e.g., 1,4-dihydroxy-2-naphthoic acid: DHNA), and tryptophan metabolites (e.g., the tryptophan photoproduct FICZ) [4]. AhR is ubiquitously expressed in vertebrate cells, including mast cells. Upon ligand binding, AhR translocates to the nucleus and initiates the transcription of target genes with promoters containing xenobiotic responsive element (XRE) consensus sequences, including AhR repressor (*AhRR*) as well as *Cyp1A1* and *Cyp1B1*, which are members of the phase I enzyme cytochrome P450 family of xenobiotic enzymes. Notably, in the past decade, there has been increasing evidence that AhR signaling plays an important role in the regulation of innate and adaptive immune responses beyond xenobiotic metabolism [5]. For instance, AhR controls mast cell differentiation and activation (e.g., FICZ enhances the IgE-mediated degranulation in mast cells) [6,7,8,9]. Interestingly, high-affinity AhR ligands are present in certain culture media. Veldhoen et al. reported that Iscove’s Modified Dulbecco’s Medium (IMDM) contains a three- to five-fold higher amount of AhR ligands (e.g., tryptophan phenylalanine) when compared to the RMPI 1640 medium [10]. As a result, IMDM promotes the in vitro differentiation of Th17 cells or dendritic cells more strongly than the RPMI 1640 medium, depending on these cells’ intrinsic AhR activity [10,11].

This study demonstrates that EC inhibits the activation of AhR signaling, and this is linked to the suppressive effects of EC on IgE-mediated mast cell activation with background AhR activity. Given that AhR signaling controls inflammation, the current finding may provide a novel mechanistic insight into how EC mediates anti-inflammatory properties. Additionally, these results suggest that IgE-mediated mast cell activation in vivo may be enhanced by endogenous AhR activity at the site of the affected tissue, such as the skin.

## 2. Results

### 2.1. EC Antagonizes AhR Activation Both In Vitro and In Vivo

Since many phytochemicals function as AhR agonists or antagonists [12], we used a previously reported in vitro bioassay system [13] to determine whether EC also has the potential to affect the AhR pathway. In brief, a murine hepatoma cell line (Hepa-1c1c7) was stably transfected with the secreted alkaline phosphatase (SEAP) gene under the control of XRE consensus sequences. Accordingly, the established sensor clone, HeXS34, secreted SEAP following stimulation with either FICZ or DHNA, both of which are AhR ligands. We found that 10 μM EC alone did not have AhR agonistic activity, but instead, a pretreatment with 10 μM EC significantly inhibited the SEAP activity induced by FICZ or DHNA in HeXS34 reporter cells (Figure 1B). FICZ- or DHNA-induced *Cyp1A1* mRNA expression was consistently inhibited by 1 or 10 μM EC in mouse bone marrow-derived mast cells (BMMCs), and this inhibition also occurred following treatment with CH-223191, a well-established AhR antagonist (Figure 1C). We confirmed that EC at concentrations of 1–25 μM did not affect the cell viability in HeXS34 reporter cells or BMMCs, based on direct cell counts and WST and annexin V assays (Appendix A). 

Exposure to AhR ligands causes AhR-expressing cells to downregulate AhR expression at the protein level through the ubiquitin/proteasome degradation pathway [14,15]. Twelve hours after FICZ exposure, AhR protein levels were reduced in unstimulated BMMCs, as previously described [7]. EC at 1 and 10 μM inhibited the FICZ-induced downregulation of AhR protein levels (Figure 1D). However, a concentration-dependent effect of EC in rescuing AhR protein levels was not observed (Figure 1D). In addition, AhR ligands alone were reported to stimulate mast cells to release IL-6 [7]. We consistently found that DHNA did induce IL-6 production by BMMCs, and this production was inhibited by EC at 10 μM (Figure 1E).

We previously showed that a single oral dose of DHNA in mice induced *CYP1A1* mRNA expression in the small intestine [16]. Pretreatment of mice with EC (10 mg/kg, p.o.) or CH-223191 inhibited DHNA-induced *CYP1A1, CYP1B1*, and *AhRR* expressions in the small intestine (Figure 1F). These results suggest that EC can inhibit the activation of AhR signaling both in vitro and in vivo.

### 2.2. EC Inhibits AhR-Mediated Potentiation of Mast Cell Activation

To explore the potential functional impact of the AhR-inhibitory activity of EC on inflammation, we used IgE-mediated allergic inflammatory reactions in BMMCs as a model since AhR controls mast cell activation [6,7,8,9]. 

Several commercially available culture media contain AhR ligands. For example, IMDM contains high amounts of AhR ligands when compared to the RMPI 1640 medium [10]. We confirmed that HeXS34 reporter cells cultured in IMDM secreted more SEAP at baseline than those cultured in the RPMI 1640 medium and that this secretion was inhibited by 10 μM EC or CH-223191 (Appendix A). Consistent with previous findings that AhR ligands potentiate IgE-mediated degranulation in mast cells [6,7,8,9], we found that BMMCs cultured for 2 weeks in IMDM showed enhanced degranulation upon IgE stimulation compared to those cultured for 2 weeks in the RPMI 1640 medium, as assessed by β-hexosaminidase (β-hex) release (Figure 2A). Notably, both EC at 10 μM and CH-223191 significantly inhibited the IgE-mediated degranulation in BMMCs cultured in the RPMI 1640 medium or IMDM, as assessed by the β-hex and CD63 expression (Figure 2B,C). The combination of EC and CH-223191 did not enhance the suppressive effects of each individual reagent on degranulation, suggesting that EC and CH-223191 might share the same target (Appendix A). In addition, both EC and CH-223191 inhibited the degranulation in mucosal-type mast cells (MMCs) differentiated in the co-culture of BMMCs with notch ligand-expressing Chinese hamster ovary cells (Figure 2D) [17]. These results suggest that EC inhibits the AhR-dependent potentiation of IgE-mediated mast cell activation due to its intrinsic AhR activity in culture medium in vitro.

Passive cutaneous anaphylaxis (PCA) reaction is a classic in vivo model of IgE-mediated mast cell degranulation in the skin of rodents. In healthy skin, AhR signaling is constitutively active and plays an important role in maintaining tissue homeostasis [18,19]. For instance, many AhR ligands, including FICZ, are present in normal skin [20], and AhR-deficient mice show severe abnormalities in keratinization and skin barrier function [21]. Thus, we hypothesized that endogenous AhR activity in the skin could affect the PCA reaction in a similar way that AhR ligand-containing culture medium affected IgE-mediated degranulation in BMMCs (Figure 2B,C). 

As expected, oral administration of EC or CH-223191 24 h prior to the induction of the PCA reaction significantly decreased ear swelling, the quantity of Evans blue dye extracted from ear tissue, and the serum MCP-1 levels compared to the vehicle treatment (Figure 2E–G). The reduction witnessed in the PCA reaction by EC or CH-223191 was associated with the suppression of constitutively expressed CYP1A1 and AhRR in the skin (Figure 2H). These results suggest that EC inhibits the AhR-dependent potentiation of IgE-mediated mast cell activation due to its intrinsic AhR activity in mouse skin in vivo.

## 3. Discussion

This study is the first to show that EC has the potential to antagonize AhR signaling. EC has been reported to have not only anti-inflammatory activity [2,3] but also antioxidant [22], anti-malaria [23], and anti-carcinoma activity [24]. Since AhR signaling plays pleiotropic roles in several biological activities that overlap with EC activities [3,4,25], the inhibition of AhR signaling by EC may underlie its diverse biological effects. This study also showed that IgE-mediated mast cell activation might occur in the context of background endogenous AhR activity in vivo, suggesting that the blockade of the AhR signaling by EC might be beneficial for allergic diseases involving IgE and mast cells.

Both EC and CH-223191 inhibited the IgE-mediated degranulation in BMMCs cultured in RPMI 1640 medium or IMDM containing AhR ligands (Figure 2A–C). These findings appear to contradict those of Sibilano et al., who showed that prolonged AhR stimulation inhibited the IgE-mediated degranulation in BMMCs [7]. This could be due to different experimental conditions. In the current study, we prepared mature BMMCs that had been cultured in RPMI 1640 medium in the presence of IL-3 for 4 weeks. Then, we replaced the RPMI 1640 medium with IMDM (or continued to use the RPMI1640 medium) and cultured BMMCs for another 2 weeks before the assays. In contrast, Sibilano et al. treated the BMMCs twice with FICZ for ~2 weeks. In any case, the current findings suggest that the presence of AhR ligands in a cell culture medium influences the mast cell activation levels as well as Th17 cell differentiation and dendritic cell maturation [10,11]. Therefore, an optimal medium selection seems necessary when studying BMMCs or MMCs in vitro. 

Our data suggest that the PCA reaction may occur with a background of endogenous AhR activation in mouse skin. The skin contains various AhR ligands, including FICZ [20]. In addition, AhR-deficient mice show severe abnormalities in keratinization and skin barrier function [21]. Thus, it is likely that persistent background AhR activity in the skin modulates the PCA reaction. 

There have recently been many reports that AhR activation can suppress inflammatory skin conditions, such as atopic dermatitis and psoriasis [26,27], and these reports seem to contradict the findings of our study. In these reports, keratinocyte-specific AhR is the key to downregulating skin inflammation [27]. Therefore, the outcome of AhR signaling in the skin may depend on the cell types in which AhR is activated or on a variety of inflammatory contexts in which many other signaling pathways, such as EGFR, mitogen-activated protein kinase, NF-κβ, β-catenin, and JAK/STAT pathways, can interfere with AhR signaling [28]. Actually, it has been reported that, while blocking AhR activation is desirable in some skin conditions, stimulating this activation is beneficial in another group of skin disorders [28].

The most important question is how EC can inhibit AhR signaling. The combination of EC and CH-223191 did not enhance the suppressive effects of each individual reagent on the IMDM-dependent increase in degranulation in BMMCs, suggesting that EC and CH-223191 might share the same target (Appendix A). CH-223191 blocked the binding of TCDD to AhR and inhibited both the TCDD-mediated nuclear translocation and the binding of AhR to DNA, thereby preventing the expression of AhR target genes [29]. Unlike the many AhR antagonists known, CH-223191, like EC, was shown to not have detectable AhR agonist-like activity [30]. Thus, it is possible that EC has the same mechanisms as CH-223191. It is, therefore, crucial to determine whether EC can bind to AhR and, if it can, to evaluate the affinity of this binding.

In summary, we propose that EC, a caffeic acid-derivative polyphenol, can inhibit the activation of AhR, an important transcriptional factor that senses and responds to various environmental stimuli and plays fundamental roles in immune regulation [3]. These findings suggest a novel mechanism underlying the anti-inflammatory activity of EC. Additionally, EC might have the potential to prevent or treat allergic diseases that involve IgE and mast cells since intrinsic AhR activity may play an important role in shaping allergic inflammation.

## 4. Materials and Methods

### 4.1. Reagents

Ethyl caffeate (EC) was purchased from Selleck, DC, USA (catalog No. S5640 Batch No. 01). FICZ, DHNA and CH-223191 were purchased from Sigma-Aldrich^®^ Merck, KGaA (Darmstadt, Germany). The RPMI1640 medium, IMDM, and MEM-a were purchased from Nacalai Tesque (Kyoto, Japan).

### 4.2. Mice

Male C57BL/6 mice, aged 8–12 weeks old, were purchased from the Jackson Laboratory (Tokyo, Japan SLC) and were synchronized for at least two weeks with a time cue of 12 h light:12 h dark settings. All animal experiments were approved by the Institutional Review Board of the University of Yamanashi.

### 4.3. SEAP Assay

HeXS34 (Hepa-1c1c7-derived, pXRE-SEAP-transfected clone no. 34) cells were kindly provided by Dr. Yao (University of Yamanashi, Chuo, Yamanashi, Japan). Briefly, the HeXS34 cells were established by transfection of the Hepa-1c1c7 cells with pXRE-SEAP, which introduces a SEAP gene under the control of two copies of the XRE consensus sequence [13]. The cells were maintained in MEM-a, supplemented with 10% FBS, 1% penicillin/streptomycin [P/S] (Nacalai Tesque, Kyoto, Japan), and 500 µg/mL of G-418 (FUJIFILM Wako, Osaka, Japan). 

To evaluate the inhibitory effects of the AhR activity by EC, HeXS34 cells (1 × 10^5^ cells/mL) were seeded in 96-well plates overnight before being treated with EC (1 or 10μM) and CH-223191 (1 or 10 μM) for 1 h, followed by stimulation of 1nM FICZ or 5μM DHNA for 12 h. The activity of SEAP in the supernatants was evaluated by a chemiluminescent method using the Great EscAPe SEAP detection kit (Clontech, Fitchburg, WI, USA) following the manufacturer’s instructions on the kit, as previously described [16]. To compare the SEAP activity in the RPMI1640 medium, MEM-α, and IMDM, HeXS34 cells were seeded in 96-well plates that were cultured in RPMI1640 medium, MEM-α, or IMDM with or without EC or CH-229191 for 24 h. 

### 4.4. Generation of Mouse Bone Marrow-Derived Mast Cells (BMMCs) and Mucosal Mast Cells (MMCs)

The BMMCs and MMCs were generated as previously described [17]. Briefly, sterile bone marrow was harvested from the mice femur and cultured in RPMI1640 supplemented with 10% FBS and 10 μg/mL of recombinant mouse IL-3 (Peprotech, Cranbury, NJ, USA) to establish the BMMCs. After 4 to 6 weeks of culture, mature mast cells were identified by using the flow cytometric detection cell-surface c-kit (CD117) and FcεRIα (purity > 95% c-kit^+^ FcεRIα^+^). For the experiments of IMDM-cultured BMMCs, 4-week-RPMI-cultured BMMCs were cultured for 2 weeks in IMDM supplemented with 10% FCS and 10 μg/mL recombinant mouse IL-3.

For the MMCs, control Chinese hamster ovary (CHO) cell lines or DLL1-expressing CHO cell lines were seeded into a culture dish and treated with 3 mg/mL of mitomycin C (Sigma-Aldrich^®^ Merck KGaA, Darmstadt, Germany) for 3 h. Two-week-cultured-BMMCs were placed at 1 × 10^6^ cells/mL onto the CHO cells’ seeded dish and co-cultured with the control or DLL1-expressing CHO cells in MEM-a supplemented with 10% FBS and 10 ng/mL recombinant mouse IL-3. After the co-culture, dead cells were removed by magnetic cell sorting using a dead cell removal kit (Miltenyi Biotec, Bergisch Gladbach, Germany), and then the c-kit-positive cells were purified by magnetic cell sorting using a magnetic microbead-conjugated anti-mouse CD117/c-kit mAb (Miltenyi Biotec, Bergisch Gladbach, Germany), according to the manufacturer’s instructions (purity > 95% c-kit^+^ FcεRIα^+^ cells).

### 4.5. FACS Staining 

The BMMCs or MMCs were incubated for 15 min with rat-anti-mouse Abs to CD16/32 (2.4 G; BD Biosciences, CA, USA) to block nonspecific binding, and then were stained with FITC-conjugated anti-mouse FcεRIα (MAR-1; eBioscience, San Diego, CA, USA) and a PE-conjugated anti-mouse c-kit Ab (2B8; BD PharMingen, CA, USA) in PBS. For some experiments, APC-conjugated anti-mouse CD63 Ab (HMa; eBioscience, San Diego, CA, USA) was used for staining. After being washed with PBS, the stained cells (live-gated based on the forward and side scatter profiles) were analyzed using quantitated BD AccuriTM C6 flow cytometry (Becton Dickinson, Franklin Lakes, NJ, USA), and the data were processed using the BD AccuriTM C6 Flow cytometry software program Ver. 1.0.264.21 (Becton Dickinson, Franklin Lakes, NJ, USA).

### 4.6. Quantitative Real-Time PCR 

The mast cell samples were collected from the BMMCs treated with EC (1 or 10 μM) or CH-223191 (1 or 10 μM) for 1 h and were then stimulated with FICZ (1 nM) or DHNA (5 μM) for 2 h. The ear samples were collected from the mice at 24 h after treatment with EC or CH-223191 (10 mg/kg, p.o.). The jejunum samples were collected from the mice pretreated with EC or CH-223191 (10 mg/kg, p.o.) and treated with DHNA (20 mg/kg p.o.) for 4 h. 

Total RNA was collected from the BMMCs or mouse tissues by using the Rneasy Mini kit (Quiagen, Hulden, Germany). The cDNA was synthesized from the RNA samples by using the ReverTra Ace^TM^ qPCR RT Master Mix with gDNA remover (TOYOBO, Osaka, Japan). A quantitative real-time PCR analysis using cDNA from the BMMCs, mouse ear, or mouse jejunum was performed using the StepOneTM real-time PCR system (Applied Biosystems, CA, USA) according to the manufacturer’s instructions, using primers and probes for mouse *Cyp1a1*, *Cyp1b1*, *AhRR*, and *GAPDH* (Applied Biosystems, Carlsbad, CA, USA), as previously described [16]. The ratio of the indicated genes to that of GAPDH was calculated, and the relative expression levels were shown.

### 4.7. Cell Viability

Cell viability was assessed using the calorimetric WST assay, direct cell counting, and Annexin V staining. For the WST assay, HeXS34 cells or BMMCs treated with the indicated concentrations of EC, CH-223191, FICZ, or DHNA for 24 h were subjected to the WST8 assay (Cell Counting Kit-8, DOJINDO LABORATORIES, Kumamoto, Japan), following the manufacturer’s instructions.

For direct viable cell counting, the BMMCs (1 × 10^6^ cells/ mL) were treated with the indicated concentrations of EC, CH-223191, FICZ, or DHNA for 24 h, and the number of BMMCs was subsequently counted by a hemacytometer. 

For apoptosis cells detection, the BMMCs (1 × 10^6^ cells/ mL) were treated with the indicated concentrations of EC and CH-223191 for 24 h and were evaluated by FACS using the FITC conjugated-Annexin V Apoptosis Detection Kit (DOJINDO LABORATORIES, Kumamoto, Japan), following the manufacturer’s instructions.

### 4.8. Western Blot Analysis

The BMMCs (1 × 10^6^ cells/mL) were treated with EC (1 or 10 μM) and CH-223191 (1 or 10 μM) for 1 h and were stimulated with FICZ (1 nM) for 12 h. The AhR protein levels were then detected by Western blot analysis. Briefly, the BMMCs were lysed in RIPA buffer (FUJIFILM Wako, Osaka, Japan) with a protease inhibitor cocktail (Merck Millipore, Burlington, MA, USA). The cell lysates were dissolved in a sample buffer containing 50 mM of dithiothreitol and bromophenol blue and then boiled for 5 min. The protein concentrations were measured on a NanoDrop ND-1000 (Thermo Fisher Scientific, Waltham MA, USA). Proteins were subjected to SDS-PAGE gels and transferred to polyvinylidene fluoride membranes. The blots were immersed in 5% milk-blocking solution for 1 hour at room temperature (RT), followed by incubation with Anti-AHR antibody (1/1000 dilution, NOVUS BIOLOGICALS LLC, Centennial, CO, USA) and anti-β-actin (13E5, 1/1000 dilution, Cell signaling technology, Danvers, MA, USA) solution overnight at 4 °C. Membranes were washed three times with TBS/T, and then incubated in an anti-rabbit IgG, HRP-linked antibody (1/2000 dilution) solution for 40 min at RT. Immunoreactive proteins were visualized using ECL Prime (GE Healthcare, Chicago, IL, USA). Quantitative analysis of Western blotting was conducted by using the Scion Image software packages ver. 4.5.0 (a free-of-charge software; the property of Scion Corporation), and relative intensities of the target proteins to β-actin were shown.

### 4.9. ELISA

The BMMCs were treated with EC (1 or 10 μM) and CH-223191 (1 or 10 μM) for 1 h, and then DHNA (50 μM) was added to the culture. After 24 h, the IL-6 concentrations in the culture supernatants were measured by using the Mouse IL-6 ELISA kit (R & D systems, Minneapolis, MN, USA). Mouse sera were collected to evaluate the MCP-1 levels in the blood at 180 min after induction of the PCA reaction, and the MCP-1 levels were measured using a Mouse MCP-1 ELISA kit (R & D systems, Minneapolis, MN, USA).

### 4.10. β-Hexosaminidase Release and Surface CD63 Expression

The β-hexosaminidase release assay was performed as previously described.^17^ Briefly, the BMMCs or MMCs were pretreated with EC or CH-223191 for 12 h. Then, the BMMCs or MMCs were incubated with 1 μg/mL of anti-DNP mouse IgE mAb overnight at 37 °C and stimulated with 1 μg/mL of anti-mouse IgE antibody for 40 min at 37 °C. The total release was obtained by adding 1% Triton buffer for 40 min. The supernatants were collected from each well and mixed with *p*-nitrophenyl-*N*-acetyl-β-D-glucosaminide to determine the enzymatic activity of the released β-hexosaminidase. After 90 min at 37 °C, the reaction was stopped by adding 0.2 M of the glycine solution and measured by using an absorption spectrometer (at 405 nm). The percentage of β-hexosaminidase release was calculated as follows:

β-hexosaminidase release (%) = OD of stimulated supernatant/OD of supernatant of Triton-lysed cells × 100.

To evaluate the surface CD63 expression, the BMMCs or MMCs were collected after induction of degranulation, as described above, and then stained with c-kit (CD117), FcεRIα, and CD63 by FACS.

### 4.11. Passive Cutaneous Anaphylaxis (PCA) Reaction

A PCA reaction was performed, as previously described [30]. Briefly, one side of the mouse ear was intradermally sensitized with mouse IgE anti-TNP (100 ng/10 µL/ear), whereas the other was intradermally administrated with PBS as a negative control. After 24 h of treatment EC or CH223191 (10 mg/kg p.o.), the mice were intravenously administrated TNP-BSA (50 μg/200 µL)/0.2% Evans blue dye mixture. The ear’s thickness was measured before the TNP-BSA challenge (0 min) and then at 15, 30, 60, 90, and 180 min. An increased ear thickness was determined by deducting the ear thickness value at 0 min from every time point of ear thickness. After 180 min, blood samples were collected to evaluate the serum MCP-1 levels by ELISA. To evaluate the Evans blue extravasation in the ear, the ears of the mice were chopped into small pieces and were mixed with the N, N-Dimethylformamide solution and incubated for 3 h at 55 °C. Supernatants were collected, and the measured absorbance values were obtained with a 650 nm filter.

The mice ear samples were collected after oral administration of EC or CH-223191 for 24 h.

### 4.12. Statistical Analysis

Data were analyzed using GraphPad Prism 8 (GraphPad Software Inc., Boston, MA, USA). The results are expressed as the mean ± SD, and the “n” numbers for each dataset are provided in the figure legends. The statistical significance was assessed by a two-tailed Student’s t-test, one-way ANOVA, followed by Tukey’s post hoc test, or two-way ANOVA, followed by Tukey’s or Dunnett’s post hoc test. *p*-values of < 0.05 were considered significant.

## Figures and Tables

**Figure 1 ijms-24-09997-f001:**
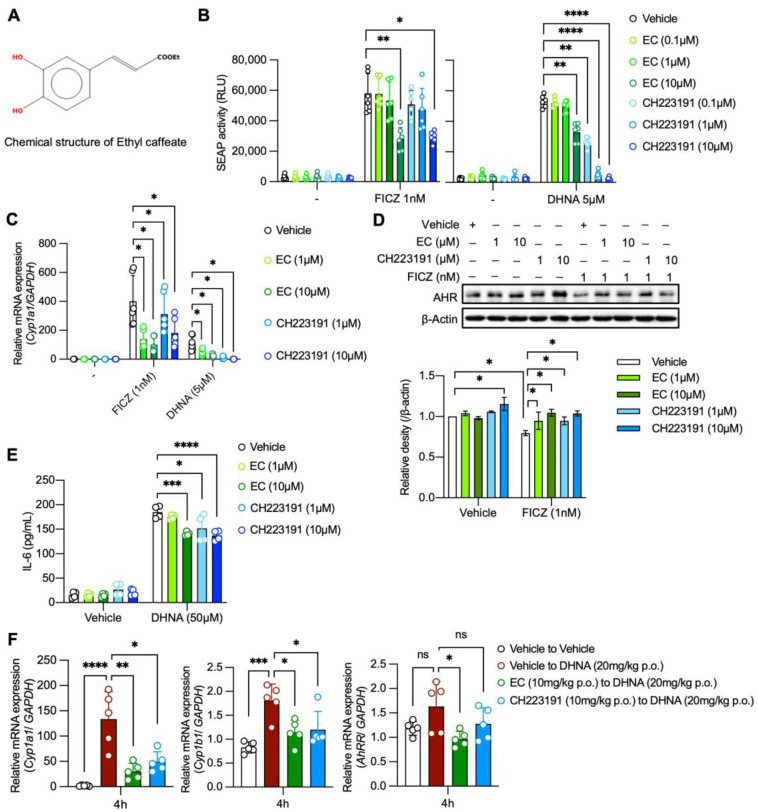
Ethyl caffeate antagonizes AhR activation. (**A**) Chemical structure of ethyl caffeate (EC). (**B**) SEAP activity induced by FICZ (**left**) or DHNA (**right**) in HeXS34 reporter cells with pretreatment of EC or CH-223191 or vehicle. Activity of SEAP in the culture supernatants was evaluated by chemiluminescence (n = 6). (**C**) *Cyp1A1* mRNA expression levels induced by FICZ or DHNA in BMMCs with pretreatment of EC or CH-223191 or vehicle (n = 6). (**D**) Upper panel: AhR protein levels after stimulations with FICZ or DHNA in BMMCs with pretreatment of EC or CH-223191 or vehicle. AhR protein levels were evaluated by Western blot analysis. Representative data are shown. Bottom panel: Quantification of AhR protein levels from Western blot analysis (n = 3). (**E**) IL-6 production induced by DHNA (50 μM) in BMMCs with pretreatment of EC or CH-223191 or vehicle evaluated by ELISA (n = 4). (**F**) *Cyp1a1, Cyp1b1,* and *AhRR* mRNA expressions in the mouse jejunum. Mice were pretreated with EC or CH-223191 or vehicle for 24 h and then challenged with DHNA (20 mg/kg p.o.). Four hours after the DHNA treatment, mice jejunum tissues were collected and evaluated for *Cyp1a1, Cyp1b1*, and *AhRR* expressions by qPCR (n = 5 per group). Mean ± SD is shown. Statistical differences were determined by one-way ANOVA with Dunnett’s post hoc test or two-way ANOVA with Tukey’s post hoc test, * *p* < 0.05, ** *p* < 0.01, *** *p* < 0.001, **** *p* < 0.0001, ns: not significant.

**Figure 2 ijms-24-09997-f002:**
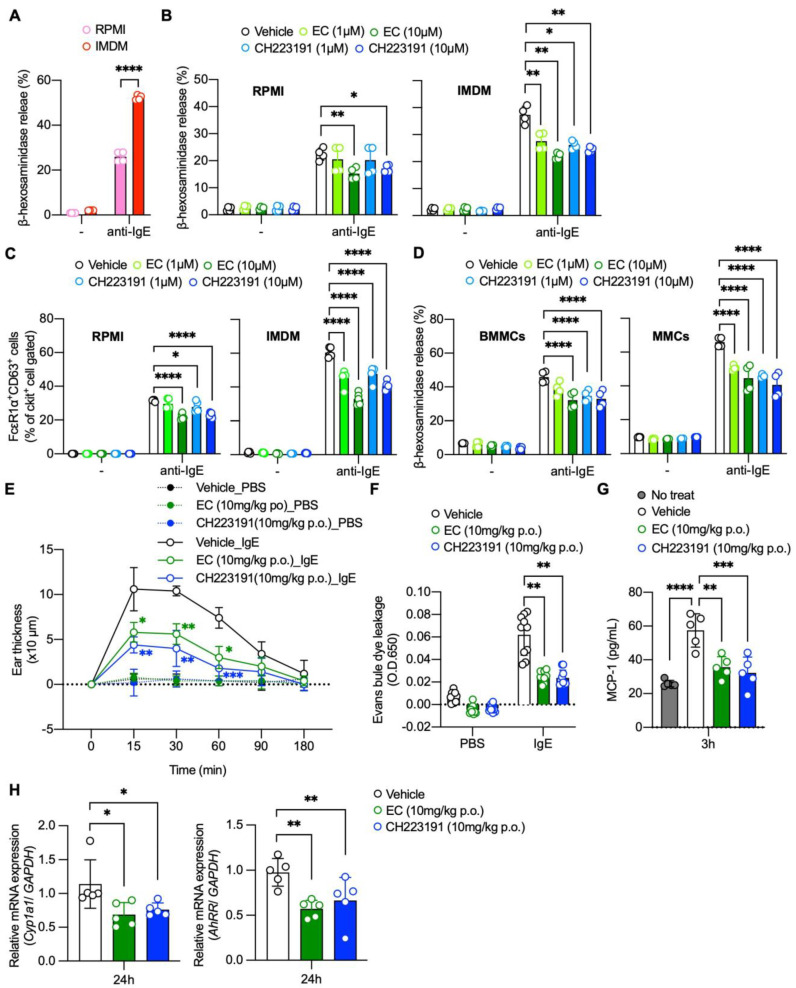
EC inhibits AhR-mediated potentiation mast cell activation. (**A**) IgE-mediated release of β-hexosaminidase in RPMI1640- or IMDM-cultured BMMCs (n = 4). (**B**) IgE-mediated release of β-hexosaminidase in RPMI1640- or IMDM-cultured BMMCs with or without pretreatment of EC or CH-223191 for 12 h (n = 4). (**C**) IgE-mediated upregulation of CD63 in RPMI1640- or IMDM-cultured BMMCs with or without pretreatment of EC or CH-223191 for 12 h evaluated by FACS (n = 4). (**D**) IgE-mediated release of β-hexosaminidase in BMMCs or MMCs with or without pretreatment of EC or CH-223191 for 12 h (n = 4). (**E**,**F**) Parameters of PCA reaction in mice pretreated with EC (10 mg/kg p.o.) or CH-223191 (10 mg/kg p.o.) or vehicle for 24 h. Ear thickness (n = 5 per group) (**E**), Evans blue extravasation (n = 10 per group) (**F**), and serum MCP-1 levels (n = 5 per group) (**G**) at 180 min after the induction of PCA reaction. (**H**) *Cyp1a1* and *AhRR* mRNA expressions in the skin of mice treated with EC (10 mg/kg p.o.) or CH-223191 (10mg/kg p.o.) or vehicle. Mice ear samples were collected 4 h after the treatment to evaluate mRNA expression for *Cyp1a1* and *AhRR* expressions by qPCR (n = 5 per group). Mean ± SD is shown. Statistical differences were determined by one-way ANOVA with Dunnett’s post hoc test or two-way ANOVA with Tukey’s post hoc test, * *p* < 0.05, ** *p* < 0.01, *** *p* < 0.001, **** *p* < 0.0001.

## Data Availability

Not applicable.

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
