# Peer review of "Ethyl Caffeate Can Inhibit Aryl Hydrocarbon Receptor (AhR) Signaling and AhR-Mediated Potentiation of Mast Cell Activation"

_ijms, 2023, doi:10.3390/ijms24129997_

Round 1

Reviewer 1 Report

The most important finding of this manuscript is that a natural product, ethyl caffeate (EC), exerts its activity on immune regulation by affecting the AhR (aryl hydrocarbon receptor) signaling pathway, at least in part.   The conclusion is drawn from the results of multiple bioassays both in vitro and in vivo, so it is solid.  Although the authors did not provide clear evidence to prove whether EC physically interacts with AhR, which lowers novelty and significance of the work, the research described here matches the scope of the special issue and its quality is suitable for publication as a communication in IJMS.

Major points:

1.  Based on Figure 1D, it is claimed that EC/CH-223191 can rescue AhR protein expression upon AhR activation.  However, this rescuing effect is not concentration-dependent, that is, the effect at 1 uM is at least equal to or even stronger than that at 10 uM.  The authors should acknowledge this observation in the text.

Minor points:

1.  In line 143, ‘Figure 2F’ should be ‘Figure 2H’.

2.  In line 127, define ‘MMCs” for mucosal-type mast cells.

3.  In line 131, define ‘PCA’.

Author Response

Reviewer 1:

The most important finding of this manuscript is that a natural product, ethyl caffeate (EC), exerts its activity on immune regulation by affecting the AhR (aryl hydrocarbon receptor) signaling pathway, at least in part. The conclusion is drawn from the results of multiple bioassays both in vitro and in vivo, so it is solid. Although the authors did not provide clear evidence to prove whether EC physically interacts with AhR, which lowers novelty and significance of the work, the research described here matches the scope of the special issue and its quality is suitable for publication as a communication in IJMS.

We thank the reviewer for the precious time carefully reviewing our manuscript and providing valuable comments. We appreciate the reviewer’s positive comments and insightful suggestions to improve our manuscript. We have carefully considered your comments and have made revisions to address the concerns raised. Please refer to our responses below for details. The revised parts of the manuscript are indicated in red.

Major points:

  1. Based on Figure 1D, it is claimed that EC/CH-223191 can rescue AhR protein expression upon AhR activation.  However, this rescuing effect is not concentration-dependent, that is, the effect at 1 uM is at least equal to or even stronger than that at 10 uM.  The authors should acknowledge this observation in the text.

Thank you for your very careful observation and suggestion. We agreed that the rescuing effect of EC was not concentration-dependent, at least at protein levels. In this revision, we added quantification results from our western blot (Figure 1D bottom panel). The result showed that the effect at 1 µM was equal to at 10 µM. Thus, we mentioned that in the results part, section 2.1 from line 84 to 86: “EC at 1 and 10 µM inhibited FICZ-induced downregulation of AhR protein levels (Figure. 1D). However, a concentration-dependent effect of EC in rescuing AhR protein levels was not observed (Figure. 1D).”

Minor points:

  1. In line 143, ‘Figure 2F’ should be ‘Figure 2H’.

We corrected “Figure 2F” to “Figure 2H” in line 146 (line number was changed because we added some sentences in response to your major comment).

  1. In line 127, define ‘MMCs” for mucosal-type mast cells.

We define “MMCs” as abbreviation for “mucosal-type mast cells” in line 129.

  1. In line 131, define ‘PCA’.

We added “Passive cutaneous anaphylaxis” before “(PCA)” to define it in line 133.

Reviewer 2 Report

The authors in this paper explore the role of ethyl caffeate on mast cell function.

The authors present a superficial collection of data to show that EC has anti-inflammatory activity.

Main criticisms:

The authors grow BMMCs with only IL-3 and no SCF. These are not necessarily mast cell like cell but more basophil like.

The differences observed with treatment with EC are very minor, for instance in IL-6 secretion. Would this be physiologically relevant? Did the authors look at other allergy associated cytokines that are more relevant like IL-4 and IL-13?

Figure 1D: the western blot data is not convincing to conclude that EC down-modulates AHR expression.

Author Response

Reviewer 2:

The authors in this paper explore the role of ethyl caffeate on mast cell function.

The authors present a superficial collection of data to show that EC has anti-inflammatory activity.

Thank you very much for taking the time to review our manuscript. We appreciate your valuable feedback and suggestions for improvement. We have taken diligent consideration of your comments and have undertaken substantial revisions to adequately address the raised concerns. Please refer to the responses below for our detailed discussion and revision regarding your concerns. The revised parts of the manuscript are indicated in red.

Main criticisms:

The authors grow BMMCs with only IL-3 and no SCF. These are not necessarily mast cell like cell but more basophil like.

Thank you for your concern. We cultured bone marrow cells to make bone marrow derived mast cells with only IL-3 using stablished culture method described in previously published articles (Please refer to Tokura T et al., Biosci Biotechnol Biochem, 2005, doi: 10.1271/bbb.69.1974 and Nakamura Y et al., J Allergy Clin Immunol. 2016, doi: 10.1016/j.jaci.2015.08.052). In support to our method, there is a recent article using the same culture method and check many markers as well as gene expression of BMMCs (please refer to Akula S et al., Cells. 2020, doi: 10.3390/cells9092118).

Moreover, in our studies, we identify mature mast cells by flow cytometric detection cell-surface c-kit (CD117) and FcεRIα. We performed experiments on cells with > 95% double positive for c-kit and FcεRIα (Methods part 4.4, line 249 to 250). In addition, basophils do not express c-kit (please refer to Xiaohong Han et al., Arch Pathol Lab Med 2008, doi: 10.5858/2008-132-813-ISOBBM). Thus, we believed that our culture method yielded mature mast cells.  

The differences observed with treatment with EC are very minor, for instance in IL-6 secretion. Would this be physiologically relevant? Did the authors look at other allergy associated cytokines that are more relevant like IL-4 and IL-13?

Thank you for your concern. In our experiments, we did not investigate the effects of EC on IL-4 and IL-13. Although these cytokines are allergy related and well-known, they are secreted by mast cells activation upon other receptors such as FcεRI, ST2, or Toll-like receptors (please refer to McLeod JJ et al., Cytokine. 2015, doi: 10.1016/j.cyto.2015.05.019). In our study, we conducted ELISA assays to investigate the potential release of IL-4 and IL-13 by mast cells upon stimulation with the AhR ligand (DHNA, 50 µM). However, we were unable to detect IL-4 in either the unstimulated or stimulated conditions. Furthermore, there was no significant difference observed in the concentration of IL-13 between the two conditions. As our study primarily focused on AhR-related aspects, we did not include these findings in our figures or supplementary data. For further reference, please consult the attached figure provided below.

To the best of our knowledge, AhR ligands can independently stimulate the release of IL-6 from mast cells (please refer to Sibilano R et al., J Immunol. 2012, doi: 10.4049/jimmunol.1200009). Therefore, we only further investigated IL-6 in our experiments with AhR ligand and EC.  

Figure 1D: the western blot data is not convincing to conclude that EC down-modulates AHR expression.

We appreciate your observation regarding the challenging interpretation of the western blot image and its depiction of the effect of EC. Taking your suggestion into careful consideration, we have included Figure 1D bottom panel in our revised manuscript. This new figure displays the quantification of our western blot results from three independent experiments. Because we used low concentration of FICZ (1 nM), AhR expression was suppressed slightly but significantly by EC at 1 and 10 µM successfully rescued the expression of AhR. We mentioned at line 85 and 86 of the Result 2.1. and line 319-321 of the Materials and Methods 4.8.

Round 2

Reviewer 2 Report

The authors have answered my questions regarding the data presented in their manuscript.